# Optical Gain in Commercial Laser Diodes

**Massimo Vanzi** [1,2]

1 Department of Electric and Electronic Engineering, University of Cagliari, 09123 Cagliari, Italy; vanzi@unica.it
2 Institute for Microelectronics and Microsystems (IMM), National Research Council of Italy, 40129 Bologna, Italy

**Abstract:** Optical gain and optical losses are separately measured in commercial laser diodes by simple analysis of spectral and electrical characteristics, and with no special specimen preparation or handling. The aim is to bring device analysis, for characterization and reliability purposes, closer to the intimate physical processes that rule over laser diode operation. Investigation includes resonating and non-resonating optical cavities.

**Keywords:** optical gain; losses; measurement; laser diode

## 1. Introduction

This paper is primarily addressed to the designers and manufacturers of products that employ commercial laser diodes. That user is likely to be an expert of laser diode theory and technology [1–35], but is not usually the manufacturer of lasers themselves. Nevertheless, users need lasers in their systems; systems for which, in turn, they are responsible to their own customers. Responsibility is found in several branches such as quality assurance and reliability, maintenance, and warranty. It is then the responsibility of the product manufacturer that fields failures return, and it is the same manufacturer that is in charge of qualifying the incoming lots for production lines. Failures, from field operation and from qualification tests, focus attention on the link between optoelectronic performance and laser physics and technology. Degradation itself can be observed, and its evolution monitored in time, in terms of change of performance, but only its correlation with the intimate underlying physical mechanism enables any kind of corrective action in the design and manufacturing process or drives the engineering of de-rating or redundancies. The lifetime of laser diodes is often the most critical item in reliability evaluation based on *part-stress*-analysis or *part-count*-analysis of complex equipment: it cannot (or should not) be taken from any database [35–38]. Failure physics should enter into the industrial process.

In this framework, the author has spent several years in reliability analysis in addition to laser diodes. Laser diodes are devices that have revealed the most surprising, unexpected, and peculiar failure mechanisms. The focus of this paper is not on those subtle physical phenomena, but on the bridge that links observation with interpretation. That bridge consists of the measurement of optical gain and loss, if made separately and experimentally, by means of simple methods.

More explicitly, optical gain is a cardinal quantity, a sort of primitive of many features of a laser diode. Its absolute measurement [39–54] would bring our observation close to the physical heart of the device: the balance between stimulated photon emission and absorption, which is essentially ruled by the properties of the active material, and their competition with optical losses, which depend on the design and quality of the optical cavity. Gain and loss figures then enter, in intermixed and encrypted forms, those familiar parameters such as threshold current and optical efficiency, whose measurement only gives an indirect insight of the fundamental phenomena that occur inside a laser diode.

Here we finally reach the initial prompt for this paper: optical gain in commercial laser diodes is that same optical gain as in any other laser diode; the difference is its measurability. Measuring gain is, indeed, a task that requires either special experimental setups

or the a priori knowledge of some technological information such as mirror reflectivity, confinement factor, internal loss coefficient, cavity length, lateral confinement technology, and spectral optical absorption. The customer of commercial devices has no easy access to that knowledge and, on the other side, no gain figures appear in the data sheets that come with the purchased devices.

In this paper, we will show that the standard set of DC measurements (current, voltage, output power, and spectrally resolved emission) available without modifying the devices and without special equipment, can provide a large amount of information on, separately, gain, absorption, and optical losses in commercial laser diodes [55–69].

An important point is that, throughout this paper, we will disregard the high injection level, keeping our investigation close or below the threshold condition. The final discussion will clarify a posteriori that this choice, in short, is due to our focus on the link among optical gain, optical loss, and laser threshold.

The structure of the paper is then as follows:

- Section 2 proposes some considerations about the absorption length for photons as a function of the level of gain. This will give an unusual derivation of the composition rule for internal and mirror losses, of the photon density inside a uniformly pumped optical cavity, and on the "opacity" of that cavity itself at low pumping. Section 2 is a useful step for Section 5.
- Section 3 summarizes the Hakki–Paoli [39,40] and the Cassidy [43,44] methods for measuring the difference between losses and gain in Fabry–Perot (FP) cavities, starting from a set of spectra acquired in the sub-threshold range of the laser. It also introduces a modified Cassidy method proposed few years ago by the author [60]. A common notation allows the comparison of the three methods.
- Section 4 introduces a second gain formula [64] and demonstrates its practical measurability, after introducing the concept of junction voltage. Both entries are shown to derive from papers dating back to the very beginning of laser diode studies. The combined use of the second gain formula and any of the formulas in Section 3 leads to the separate measurement of the total loss coefficient, of gain for each frequency in a spectrum, and also of the upper and lower limits for the gain. The lower limit for gain gives the optical absorption of the unpumped material.
- Section 5 extends the method of the coupled gain equations to the non-FP cavities [66]. It takes advantage of the results in Section 2, and, referring to one more formulation from the foundation papers on laser diodes, proposes an alternative formula to replace those of Section 2, suitable for gain calculation when spectra do not show any modulation caused by optical resonances.
- Section 6 summarizes the link between reliability and the proposed data measurement and interpretation methods.

References have been organized to sort their list in thematic blocks: foundation papers on laser diodes, cardinal books on laser diodes and electron devices, specific papers on gain and, finally, the author's contributions. This has been obtained by forcing somewhat the first citations from this introduction to the thematic block, in order to keep that order, and then specifically note each of them where relevant along the text.

## 2. Gain, Losses, and Gradients

Optical gain $g$ is defined, in analogy and competition with optical loss $\alpha$, as a spectral coefficient that, for a given photon energy $h\nu$ and at a defined level of injection, measures the percent spatial rate of change of the photon density $\phi_\nu$, so that the joint action of $g$ and $\alpha$ is

$$\frac{1}{\varphi_\nu} \frac{\partial \varphi_\nu}{\partial x} = g - \alpha \qquad (1)$$

Based on long teaching experience, this definition is prone to many misunderstandings.

First of all, despite its name, gain $g$ itself can *reduce* the photon density. It includes indeed both stimulated emission and absorption, and it can range between positive and

negative values. It does *not* include the contribution of spontaneous emission. The concept of *transparency*, $g = 0$, accordingly defines the situation for which stimulated emission exactly balances absorption, which is different from the balance between gain and losses that marks the trigger for laser action. This also means that photon losses due to creation of electron-hole pairs, that is, absorption, do not enter the loss coefficient $\alpha$.

The loss coefficient $\alpha$ includes other kinds of photon consumption, as from interaction with lattice defects, phonons, and intra-band transitions, together with all the escape possibilities from the optical cavity.

The last consideration implies that escape phenomena should not only include, as usual, photon losses from the mirror facets, but also photon escape towards the confinement layers and side losses related to the lateral confinement technology of the devices. This conflicts with the custom of introducing the confinement factor $\Gamma$ for the losses towards the surrounding layers and of neglecting the lateral losses. Here we will follow the formulation proposed in Appendix B of ref. [67], which collects all losses, apart from optical absorption (that is embedded into gain $g$), in the unique term $\alpha$. The same reference indicates how to recover the usual formulation with the confinement factor.

Moreover, both gain $g$ and loss $\alpha$ are *spectral* functions, as is the photon density $\phi_\nu$. Equation (1) is *not a wave equation*. It deals indeed with photons as particles, which can be created or destroyed and can propagate, but cannot interfere.

This is another crucial point. The origin of the concept of optical gain itself follows from Einstein's intuition about the stimulated emission within a framework of quantum energy exchanges [1,2]. The extension of his considerations to the non-equilibrium steady state for an infinite uniform domain leads to different forms of rate equations [15,34,67], which are the foundation for laser diode theory. Despite their different formulations, all rate equations have a constraint: when calculated at equilibrium, they must lead to a photon density $\phi_{0\nu}$ that, multiplied by the photon energy $h\nu$, coincides with the black body distribution of spectral energy density.

In such rate equation approaches, wave aspects do not appear. This has a strong impact on the concept of losses, which naturally include the concept of reflectivity from the facets of the optical cavity. Resonant wavelengths are more strongly reflected than any other wavelength in the emission spectrum, and reflectivity and losses coefficients should be modulated as well. This is not the custom when dealing with laser diodes, and the paper by Cassidy published in this same Special Issue [53] becomes a recommended reading for meditation on this point.

Even accepting the particle nature of Equation (1), another puzzling point is the hypothesis of uniform photon distribution, which makes the left-hand side null, despite the right-hand side gain changing with injection. It is not a trivial point, because of its implications for some gain measurement methods proposed in the literature.

## 2.1. A Continuity Equation for Photons

In order to comment on this point, an unusual approach [67] can be useful for readers familiar with solid-state electron devices, and in particular with the continuity equation for minority carriers [12]. It starts from the consideration that the solution of Equation (1)

$$\varphi_\nu(x) = \varphi_\nu(0)e^{-(\alpha-g)x} \tag{2}$$

identifies the term $\ell = 1/(\alpha - g)$ as the characteristic length for an exponential decay, that is analogous to the diffusion length for minority carriers. It is then possible to continue with the analogy and build up a continuity equation for photons that describes the time rate of change of their density. There will be then a photon generation term, identified in the spontaneous emission rate $R_{sp}$, and a net loss rate $c(\alpha - g)\phi_\nu$. The last is analogous to the recombination term, which is proportional to the photon density $\phi_\nu$ through the time constant $1/c(\alpha - g)$ with $c$ the speed of light in the active material (let us here neglect, for simplicity, the role of the equilibrium photon density that is considered in ref. [67]). No drift terms will enter this equation.

The analog of the *diffusion* term, in turn, can be obtained if we consider that photons do not survive multiple absorption or loss events before disappearing, and then their mean free path can be identified with that same characteristic length $\ell$ as before. The quantity $\ell c = c/(\alpha - g)$ then plays the role of a *diffusion coefficient*. This leads to

$$\frac{\partial \varphi_\nu}{\partial t} = \frac{c}{\alpha - g} \frac{\partial^2 \varphi_\nu}{\partial x^2} + R_{sp} - c(\alpha - g)\varphi_\nu \tag{3}$$

Equation (3) is then the continuity equation for photons that includes possible gradients for the photon density $\phi_\nu$.

For the steady state and uniform values of $\alpha$ and $g$, Equation (3) has the general solution

$$\varphi_\nu = \frac{R_{sp}}{c(\alpha - g)} + K_1 \exp[(\alpha - g)x] + K_2 \exp[-(\alpha - g)x] \tag{4}$$

where the constants $K_1$ and $K_2$ must be defined by the boundary conditions. The following exact solutions for the infinite, semi-infinite, and finite domains progressively approach the real cases.

### 2.2. Solution for the Infinite Domain: Emitted Optical Power and Feed Current

The infinite domain has no boundaries, so all losses are internal. For this reason, and in preparation of the next cases of bounded domains, let us simply add the suffix *i* to the loss coefficient.

The solution for the infinite domain, requiring $K_1 = K_2 = 0$, then gives a uniform photon density at any injection level:

$$\varphi_\nu = \frac{R_{sp}}{c(\alpha_i - g)} \tag{5}$$

It states that gain, no matter the range of its potential values, will never exceed losses, because as $g$ approaches $\alpha_i$, the photon density increases unlimitedly, as well as the energy required for their creation. It is a dramatically sharp assertion that will find clear evidence in experiments.

Note 1: the dimensions of $R_{sp}$ are [cm$^{-3}$], despite its name of rate. The reason is that we need to multiply both sides of Equation (5) by some frequency interval $d\nu$ to get a density (cm$^{-3}$) on the left side. On the right side the resulting term $\frac{d\nu}{c(\alpha_i - g)}$ is then adimensional, and $R_{sp}$, which is indeed a time rate per unit frequency, has the dimensions of a simple density.

Note 2: Equation (5) is in complete agreement with the results of a rate equation for the uniform infinite domain as given in ref. [67]. In that case, the explicit expression is given for photon emission and absorption rates in terms of electron-hole recombination or creation, respectively.

It is interesting to observe that multiplying Equation (5) by $c\alpha_i$ and comparing the result with Equation (3), one gets the *total loss rate* at optical frequency $\nu$

$$R_{loss} = c\alpha_i \varphi_\nu = \frac{\alpha_i}{\alpha_i - g} R_{sp} \tag{6}$$

If then we multiply Equation (6) by the photon energy $h\nu$ and the volume *Vol* of the active region, we have an expression of the emitted optical power at optical frequency $\nu$ per frequency interval of the active region.

$$P_\nu = Vol \cdot h\nu \frac{\alpha_i}{\alpha_i - g} R_{sp} \tag{7}$$

If now we integrate Equation (7) over all frequencies, we get the total optical power $P_{TOT}$ leaving the device

$$P_{TOT} = \int P_\nu d\nu = Vol \int h\nu \frac{\alpha_i}{\alpha_i - g} R_{sp} d\nu \tag{8}$$

In the same way, if we multiply Equation (6) by the electron charge $q$ and the volume *Vol*, we get the current $I_{ph}$ that is that part of the total current *I* that is converted into light

$$I_{ph} = q \cdot Vol \int \frac{\alpha_i}{\alpha_i - g} R_{sp} d\nu \tag{9}$$

The set of Equation (5) will be decisive for measuring gain in non-resonating cavities, as reported in the final section of this paper.

*2.3. Solution for the Semi-Infinite Domain: Mirror Losses and Gain-Dependent Opacity*

For a semi-infinite domain, say $x \geq 0$ and then $K_1 = 0$, the solution

$$\varphi_\nu = \frac{R_{sp}}{c(\alpha_i - g)} + K_2 \exp[-(\alpha_i - g)x] \tag{10}$$

must fulfill a boundary condition that gives accounts for photon loss from the $x = 0$ boundary. That condition resembles the case of surface recombination for minority carriers in a semi-infinite semiconductor domain pervaded by a uniform generation rate [29]. It requires that photons leave the boundary with a flux that introduces a concentration gradient that is in turn proportional to the surface density. Let $\alpha_m$ be the proportionality constant between the surface density and the gradient, which has necessarily the dimensions of a loss coefficient:

$$\frac{\partial \varphi_\nu}{\partial x}\bigg|_{x=0} = \alpha_m \varphi_\nu(0) \tag{11}$$

We will first focus on the resulting expression for the *observable* outgoing flux:

$$\frac{\partial \varphi_\nu}{\partial x}\bigg|_{x=0} = \alpha_m \varphi_\nu(0) = \alpha_m \frac{R_{sp}}{c(\alpha_i + \alpha_m - g)} \tag{12}$$

If we multiply Equation (12) by the speed of light $c$ and compare with Equation (6), we get the loss rate from the unique boundary of the semi-infinite domain

$$c\alpha_m \varphi_\nu(0) = \frac{\alpha_m}{\alpha_i + \alpha_m - g} R_{sp} \tag{13}$$

The emission, observed from outside, *looks* as a fraction $\alpha_m$ of the internal photon density of an infinite domain (Equation (5)), whose losses are now the sum of internal and mirror losses

$$\alpha_T = \alpha_i + \alpha_m \tag{14}$$

The complete solution tells another story:

$$\varphi_\nu(x) = \frac{R_{sp}}{c(\alpha_i - g)} \left[1 - \frac{\alpha_m}{\alpha_i + \alpha_m - g} \exp[-(\alpha_i - g)x]\right] \tag{15}$$

Equation (15) describes a non-uniform photon distribution that, at a distance larger than $\ell = \frac{1}{\alpha_i - g}$ from the boundary, does no more *feel* the boundary itself, and behaves as for the infinite domain, with only internal losses $\alpha$. This means that the outgoing photons only come from a surface region with extension $\ell$. Photons generated at a larger distance from the surface will be absorbed before reaching the boundary.

There is nice qualitative evidence of the last conclusion, looking at a couple of sub-threshold spectra of a DFB laser diode with (nearly) non-reflecting facets (Figure 1). Despite

the DFB structure that is present, the Bragg grating does not cause the expected resonance peak at the lowest injection level (curve A). This points out that the measured light emission comes from a layer close to the emission facet that is too thin for sustaining interference among the multiple Bragg-scattered optical beams inside it, and that this emitted light has no awareness of what happens deep inside the optical cavity.

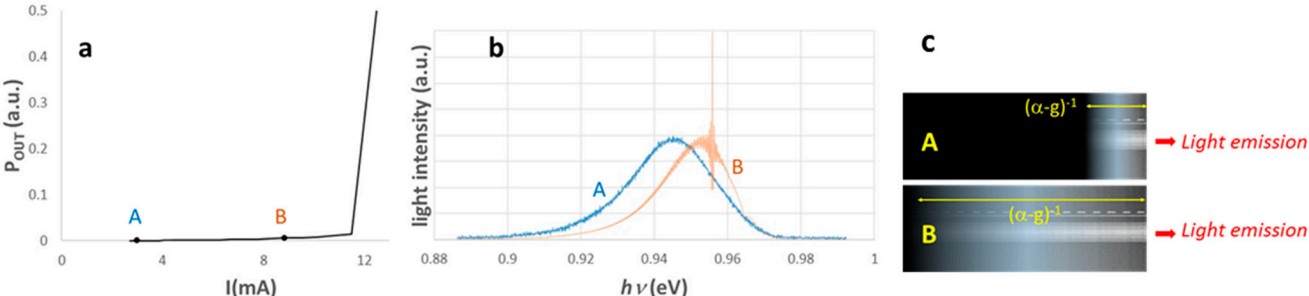

**Figure 1.** (**a**) Power-current curve of a DFB laser emitting at 1300 nm; (**b**) normalized spectra measured at 3 mA (A) and 9 mA (B) in the sub-threshold range; (**c**) graphical representation of cases A and B based on a real TEM cross-sectional image of a MQW DFB laser diode. In case A the inner part of the optical cavity does not contribute to the measured light emission. In case B the periodic grating is involved enough to cause the interference peak observed in the spectrum.

### 2.4. Considerations about the Finite Domain

Finally, it is possible to consider also a finite domain, and set different loss rates at the two boundaries of that domain. The physical meaning of the previous result will hold: if the domain has a length $L$, no photons will be able to travel the full length until $\frac{1}{\alpha_i - g} \approx L$. In particular, if $\frac{1}{\alpha_i - g} << L$ no reflected flux will interfere with the incoming one. This should be taken into account when considering non-resonating optical cavities, as in ref. [54], when the proposed analysis stands on the coherent superposition of waves travelling the whole length of the optical cavity.

The last consideration shifts the focus to resonating Fabry–Perot cavities, for which gain measurement methods rely on spectrum modulation by interference of multiplied reflected beams. Should a situation as case A in Figure 1 hold, none of those methods would work.

This limits the applicability of the most popular methods for gain measurement to injection levels high enough to allow resonances to modulate the emission spectrum. Figure 2 plots, not on the same scale, three sub-threshold spectra from a DFB laser diode with surviving FP resonances. It results that several parts in each spectrum do not display any modulation.

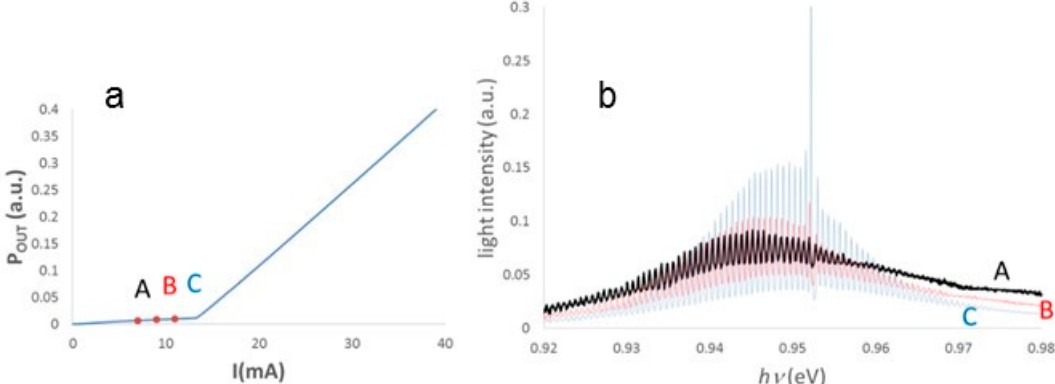

**Figure 2.** (**a**) Power-current curve of a DFB laser emitting at 1300 nm also with FP resonances; (**b**) spectra measured at 7 mA (A), 9 mA (B), and 11 mA (C), each rescaled in order to appreciate the ratio between baseline and modulation. For all spectra, no resonances appear in the low-intensity range at high photon energy.

When applicable, however, methods based on interference modulation are quite effective.

## 3. Resonating Fabry–Perot Cavity

The leading approach for measurement of gain in a laser diode with a Fabry–Perot optical cavity is the Hakki–Paoli method [40,41], which considers a plane wave $F$, with wavenumber $k = \frac{2\pi\nu}{c}$, propagating in a lossy medium, and asks its squared modulus to behave as the photon density in Equation (2):

$$
\begin{aligned}
F(x) &= F_0 e^{-\left(\frac{\alpha-g}{2}\right)x} e^{ikx} \\
|F|^2 &= |F_0|^2 e^{-(\alpha-g)x} \equiv \varphi_\nu(x)
\end{aligned}
\tag{16}
$$

It is important to realize that Equation (16) does not come from the solution of a wave equation, but from the adjustment of a true solution, the plane wave, to the decay rule of Equation (2).

The superposition of all possible waves after different round cycles of length $2L$ leads to the formula for the expected intensity spectrum from a resonating cavity

$$
\begin{cases}
|F|^2 = \dfrac{|F_0|^2}{1+\rho^2-2\rho\cos(2kL)} \\
\rho = \exp(-(\alpha_\mathrm{T}-g)L) \\
\alpha_\mathrm{T} = \alpha_i + \dfrac{1}{2L}\ln\left(\dfrac{1}{R_1 R_2}\right)
\end{cases}
\tag{17}
$$

where the loss coefficient $\alpha_\mathrm{T}$ includes all losses (apart, once again, from absorption, which is embedded in $g$), that in the third line appear separated in the internal loss coefficient $\alpha_i$ and the loss from the two mirror facets of the optical cavity, whose reflectivities are $R_1$ and $R_2$.

Here the original Hakki–Paoli method measures the envelopes $|F|^2_{MAX}$ of maxima and $|F|^2_{\min}$ of minima in Equation (17) from real spectra and calculates the quantity $\rho$ from the ratio $|F|^2_{MAX}/|F|^2_{\min}$ at different injection levels.

Integral methods, on the contrary, follow the roadmap indicated by Cassidy [43] and numerically calculate the integral of Equation (17) over one spectral range, which means a moving average over one period of the cosine function. Let us define it $|F|^2_+$. The original method of Cassidy then calculates the ratio $|F|^2_+/|F|^2_{\min}$, and gets the same quantity $\rho$ as for the Hakki–Paoli method. The advantage in this case is that the smoothing of maxima introduced by the experimental equipment is no more a concern, so that extreme resolution is no longer required for gain measurements.

An alternative way, the Vanzi method [65], is to replace $|F|^2_{\min}$ in the Cassidy method with the reciprocal of the moving average of $1/|F|^2$, that is, in turn, the reciprocal of Equation (17). Indicating this last spectral function with $|F|^2_-$, it is now the ratio $|F|^2_+/|F|^2_-$ that leads to $\rho$.

Figure 3 plots $|F|^2_{MAX}, |F|^2_{\min}, |F|^2_+, |F|^2_-$ together with the single-pass spectrum $|F_0|^2$ onto an experimental sub-threshold spectrum, and Equation (18) gives the mathematical form of the first four functions.

$$
\begin{cases}
|F|^2_{MAX} = \dfrac{|F_0|^2}{(1-\rho)^2} \\
|F|^2_{\min} = \dfrac{|F_0|^2}{(1+\rho)^2} \\
|F|^2_+ = \dfrac{|F_0|^2}{1-\rho^2} \\
|F|^2_- = \dfrac{|F_0|^2}{1+\rho^2}
\end{cases}
\qquad
\begin{cases}
\dfrac{|F|^2_{MAX}}{|F|^2_{\min}} = \left(\dfrac{1+\rho}{1-\rho}\right)^2 \quad \text{Hakki-Paoli} \\[2mm]
\dfrac{|F|^2_+}{|F|^2_{\min}} = \dfrac{1+\rho}{1-\rho} \quad \text{Cassidy} \\[2mm]
\dfrac{|F|^2_+}{|F|^2_-} = \dfrac{1+\rho^2}{1-\rho^2} \quad \text{Vanzi}
\end{cases}
\tag{18}
$$

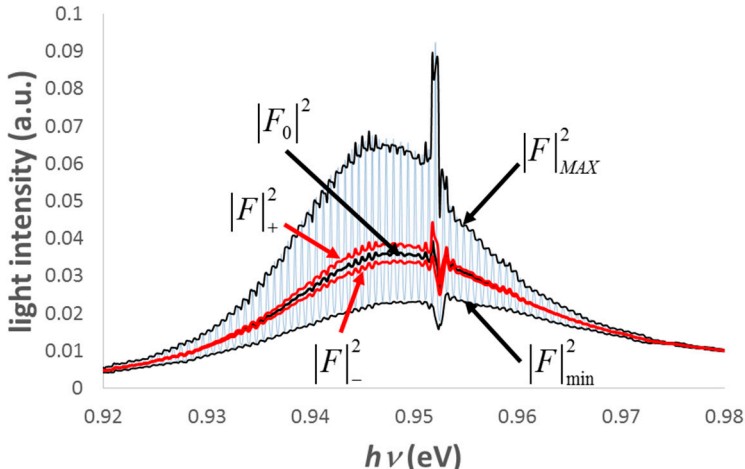

**Figure 3.** An experimental spectrum from a DFB laser diode with a Fabry–Perot cavity, with the relevant curves for the Hakki–Paoli, Cassidy, or Vanzi methods. The mathematical expression of each curve is given in Equation (18), and follows the definition of $|F|^2$ in Equation (17). All curves are calculated from experimental data. The three methods share the same approach of calculating the ratio of a pair of those curves, in order to eliminate the $|F_0|^2$ term. They differ from each other in the choice of the specific pair of curves. The benefits of replacing $|F|^2_{MAX}$ in the Hakki–Paoli method with the function $|F|^2_+$ is one of the valuable points of the Cassidy method.

In any case, the result is the measurement of $\rho = \exp(-(\alpha_T - g)L)$ for each available mode of the optical cavity by simple algebraic manipulation of the values of the measured ratios.

A by-product of all the above methods is the possibility to reconstruct the single-pass spectral intensity $|F_0|^2$, which will be important for calibrating gain measurement methods in non-resonating cavities.

It should be kept in mind that all functions $F$, and then also the function $\rho$, depend on the injection level, and then change together with spectra when the laser current is changed.

One additional positive point is that the Hakki–Paoli method applies to DFB cavities, provided no FP resonances are present at all or, in mixed cases as in Figures 2 and 3, when the DFB peak coincides with one of the FP peaks [60].

Two issues, on the contrary, are of relevant concern:

(1) Gain figures can be obtained from spectral measurements only when resonances are present. This excludes, in any case, low intensity spectra, as for low injection, or non-modulated sides of spectra as the high-energy tails in Figure 2, or the whole range of frequencies different from the DFB peak in Figure 1;

(2) Resonance-based gain measurements, when applicable, do not decouple $g$ from $\alpha_T$. In mathematical terms, solutions for $g$ alone are available only when the single-pass intensity $|F_0|^2$ in Equation (18) can be ignored.

## 4. Extending the Hakki–Paoli Method

Let us face the second point: decoupling gain from losses.

Two steps lead to the construction of an independent formula for gain that, once coupled with any other spectral measurement of light emission (even with no resonances), will lead to absolute gain and loss measurement.

The first step, proposed by Paoli and Barnes in 1976 [39], simply states that the difference $\phi_n - \phi_p$ between the quasi-Fermi levels inside the active region is *measurable* by means of the voltage $V$ and the current $I$ applied to the laser diode.

$$\varphi_n - \varphi_p = qV_J = q(V - R_S I) \tag{19}$$

Here, apart from the electron charge $q$, only one parameter enters into play: the series resistance $R_S$.

Equation (19) assumes that the voltage drops $V_J$ (junction voltage) at the ideal diode and the external bias only differ by a series ohmic contribution (Figure 4).

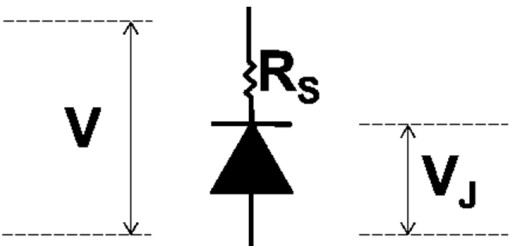

**Figure 4.** The equivalent circuit for a real laser diode implied by Equation (19).

If such simple representation holds, one expects that, as soon as the device reaches its threshold for laser emission and the quasi-Fermi levels clamp, the junction voltage $V_J$ accordingly freezes at a constant value $V_{th}$.

The calculation of $R_S$ from experimental data is then a simple task. If one, indeed, evaluates the differential form of Equation (19), one gets an expression, Equation (20), that directly gives $R_S$ when the separation of the quasi-Fermi levels $qV_J = \phi_n - \phi_p$ clamps and then the derivative is null, that is, when the forward current reaches and exceeds the threshold for laser action.

$$\frac{dV}{dI} = R_S + \frac{dV_J}{dI} \tag{20}$$

Clamping of quasi-Fermi levels is one of the most direct proofs of population inversion in a semiconductor [28], which experiments readily confirm: the plot of $dV/dI$ displays (Figure 5) a sharp transition when the current $I$ reaches the threshold value $I_{th}$ and the curve flattens at a constant value that is the measurement of $R_S$.

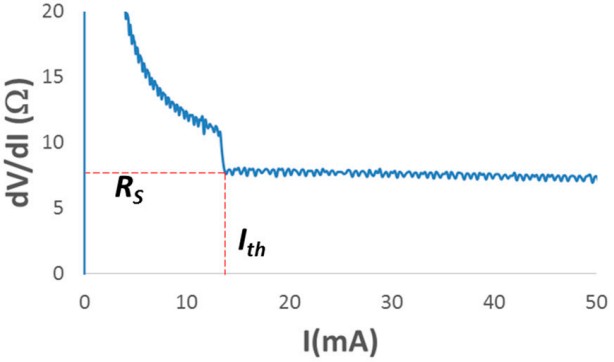

**Figure 5.** Plot of the $dV/dI$ values for the same laser diode of Figure 2. The coordinates of the sharp transition individuate the threshold current $I_{th}$ and the series resistance $R_S$. Note: The small decrease of the $dV/dI$ curve after the threshold is due to geometric effects (progressive widening, after threshold, of the area of the inverted region), analyzed and explained in ref. [67]. On a practical note, it is the $R_S$ value measured at threshold to be considered.

From the knowledge of $R_S$ one gets the direct plot (Figure 6) of the internal voltage $V_J$, by means of Equation (19), that indeed clamps as theoretically expected.

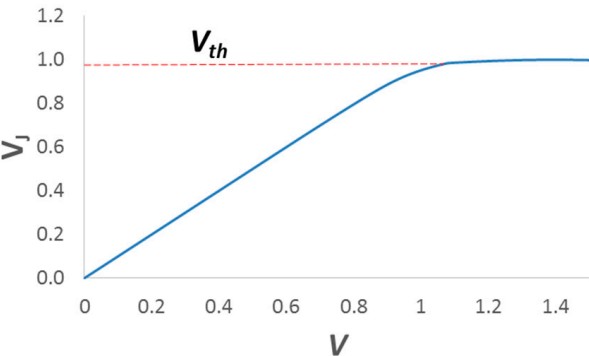

**Figure 6.** Plot of the internal voltage $V_J$ as a function of the applied voltage $V$. Internal voltage saturates at the threshold voltage $V_{th}$.

The second step goes back to 1964, when Lasher and Stern published their formulas for optical gain in a laser diode [15]. Those formulas have been considered for a long time a sound theoretical insight into laser physics, not suitable for practical measurement.

It is the statement, expressed by Equation (19), that we can *measure* the difference $\phi_n - \phi_p$ between the quasi-Fermi levels that allows us to transform the Lasher and Stern formulas into an extremely simple and manageable relationship [60,67], given in Equation (21):

$$g\left(h\nu, qV_J\right) = g_m(h\nu)\frac{1-e^{\frac{h\nu-qV_J}{2kT}}}{1+e^{\frac{h\nu-qV_J}{2kT}}}$$

$$g_m = \frac{BN_\nu^2}{c} \tag{21}$$

In this expression gain $g$ is expressed as a function of the photon energy $h\nu$, of the junction voltage $V_J$, and of a spectral coefficient $g_m$ that also defines the range $-g_m{:}g_m$ of the possible gain values. The coefficient $g_m$ collects three more fundamental quantities: the Einstein coefficient for stimulated transitions $B$, the joint density of states $N_\nu^2$ for radiative transitions at frequency $\nu$, and the speed of light $c$ in the optically active material.

Under the spectral point of view, $B$ and $c$ are both slow functions of the photon frequency, or energy. On the contrary, $N_\nu^2$ gives account for the absence of possible transitions at energies lower than the bandgap. For instance, in an ideal quantum well $N_\nu^2$ should be a step function centered at the bandgap energy $E_g$.

Figure 7 assumes, for graphical purposes, a smoothed step function to represent $g_m(h\nu)$. This corresponds to the introduction of some line shape broadening, as discussed in Section 4.3.2 of ref. [34]. For our illustrative case, a Lorentzian broadening has been assumed for plots. In practical cases, we will pretend to also get $g_m$ from measurements.

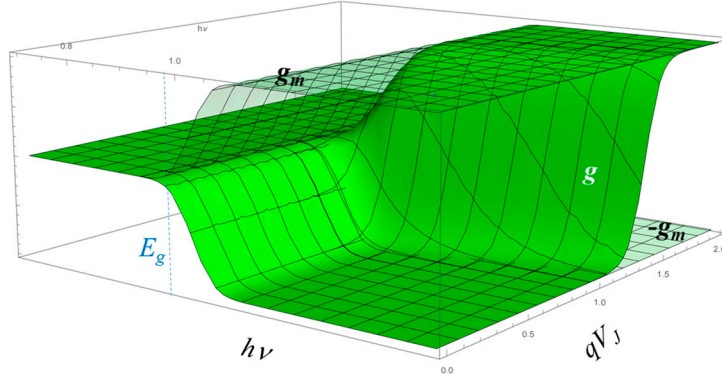

**Figure 7.** Plot of the surface $g\left(h\nu, qV_J\right)$ from Equation (21) assuming a smoothed step function to represent $g_m(h\nu)$.

The surface $g\left(h\nu, qV_J\right)$ in Figure 7 is worth some consideration. The curves, drawn at constant $qV_J$, should be recognized as the familiar gain curves as functions of the frequency $\nu$ (in our plots and formulas, the photon energy $h\nu$), as reported even in textbooks (in particular, in chapter 4.6 of ref. [34], where the abscissa is the photon wavelength instead of its frequency or energy).

Less usual are the gain curves at constant $qV_J$, at fixed photon energy $h\nu$. They show how gain ranges between a maximum and minimum, and that the two extremals have the same absolute value $g_m$ and opposite sign.

The minimum-$g_m$ corresponds to null bias $V_J$, which means null injection: the active material is essentially an absorbing medium and the gain curve becomes the absorption curve.

The maximum $g_m$ is much less intuitive, and can even be misleading. Let us first point out what the maximum gain is *not*. It is *not* the clamp value at which gain exactly equals losses, which also is the condition for achieving the laser regime. That condition, that appears from Equation (4) and in all its solutions, states that when $g = \alpha_T$, the photon density becomes infinite. It is then a physical constraint that prevents $g$ from further increasing: it would require infinite energy. On the contrary, $+g_m$ is the mathematical upper limit of the function $g$ given in Equation (21). It corresponds to an infinite separation of the quasi-Fermi levels, that describes a *completely inverted* semiconductor material (conduction band totally filled, valence band totally empty). The discriminant between the possibility or not of achieving the laser regime is the relative values of $g_m$ and $\alpha_T$. If total losses $\alpha_T$ were higher than $g_m$, the diode could be biased at any bias value, and any current could be fed into it, but that diode would never become a laser: gain would never equal losses, and then the device will remain a light emitting diode. This agrees with experimental evidence in Section 4.1.

The existence of symmetric ceiling and floor for gain is clearly pointed out also in ref. [33], Section 11.4.1. The comment to the Lasher and Stern formula (Equation (21) in this paper) is indeed (after adjusting symbols to this paper): "the format of the gain equation indicates that the gain coefficient must be between two limits.

If $V_J = 0$, then $g(\nu) = -g_m(\nu)$ the absorption coefficient without pumping.

If $V_J \to \infty$, then $g(\nu) = g_m(\nu)$, a completely inverted semiconductor.

What was absorption becomes gain".

Such a situation is often forgotten in practical applications, mainly because of two factors: the use of empirical functions for correlating gain with injection, and the custom to indicate injection by means of the laser current $I$. As an alternative to current $I$, carrier density may be used. This is a slippery point: current $I$ can increase without limits in a laser diode, whereas carrier density clamps at threshold, as shown in Figure 6 by the experimental measurement of the separation of the quasi-Fermi levels (Equation (19)). In the following, it will be shown what that physical clamp will cause on the mathematical surface of Figure 7). Referring gain $g$ to current $I$ leads to empirical formulas developed for fitting experimental data close to the $g = 0$ condition, disregarding a gain ceiling at $g_m$. A nice summary of the situation is given by Coldren [34] in Chapter 4.6, where various forms of empirical formulas are proposed for gain as a function or of current or of carrier density. None of those formulas has a mathematical upper limit. Details have been discussed by the author in ref. [59], where a new formula is proposed, derived from theory. The intriguing point is that such formulae and the empirical ones have the same power expansion at $g \approx 0$, which justifies the good fit of experimental data in practical cases. A discussion about the gain-current different formulas is reported in Appendix A.

The relevance of a ceiling for gain comes into evidence when considering degradations. Aging of a laser diode often implies an increase in internal or surface losses. This shifts the condition for lasing at higher and higher injection levels, up until the thresholds disappears, when $\alpha_T \geq g_m$.

It follows that $g_m$ itself becomes an interesting parameter to investigate, together with the loss coefficient $\alpha_T$ and gain $g$.

Equation (21) only differs by an approximation in its denominator from the Lasher and Stern formulas. In practice, Equation (21) assumes symmetric bands, so that the product $f_e(1 - f_h)$ of the distribution functions for electrons at energy $E_e$ and holes at energy $E_h$ is itself a simple function of the differences $h\nu = E_e - E_h$ and $\phi_n - \phi_p = qV_J$ as stated by Equation (19):

$$f_e(1 - f_h) = \frac{1}{\exp\left(\frac{E_e - \varphi_n}{kT}\right) + 1} \frac{1}{\exp\left(\frac{\varphi_p - E_h}{kT}\right) + 1} \approx \frac{1}{\left[\exp\left(\frac{h\nu - qV}{2kT}\right) + 1\right]^2} \tag{22}$$

Such approximation is of some relevance only at very low injection levels, when the device essentially behaves as an optical absorber, as discussed in the quoted reference [67].

Going back to the utility of Equation (21), we see that it is not sufficient by itself to measure gain. It indeed does not decouple gain $g$ from the absorption function $g_m$, as any form in Equation (18) does not separate $g$ from $\alpha_T$. However, the *joint* use of the two on at least two spectra, measured at different injection levels, leads to the relevant result of separately measuring $g$ for each spectrum together with the common functions $g_m$ and $\alpha_T$.

Let us indeed assume that a number $n$ of spectra has been measured at laser currents $I_i$, $I = 1, 2, \ldots n$.

By applying any of the methods in Equation (18) we have a corresponding set of measured functions $\rho(\nu)_i$, which we more comfortably transform into the adimensional spectral functions $S(\nu)_i = \ln(1/\rho(\nu)_i)$:

$$S(\nu)_i = (\alpha_T(\nu) - g(\nu)_i)L \tag{23}$$

At the same time, from the $I(V)$ data and the $V_J(V)$ measurements provided by Equation (19) after the measurement of $R_S$, we have the corresponding set of $V_{J1}, V_{J2}, \ldots V_{Jn}$, values that lead, by means of Equation (21), to the other set of measured adimensional spectral functions $H_i$:

$$H(\nu)_i = \frac{g(\nu)_i}{g_m(\nu)} \tag{24}$$

For each pair $S_i$, $H_i$, corresponding to the same $i$th spectrum, we can eliminate $g_i$ to get

$$\begin{cases} S(\nu)_1 = L\alpha_T(\nu) - Lg_m(\nu)H(\nu)_1 \\ S(\nu)_2 = L\alpha_T(\nu) - Lg_m(\nu)H(\nu)_2 \\ \ldots\ldots\ldots \\ S(\nu)_n = L\alpha_T(\nu) - Lg_m(\nu)H(\nu)_n \end{cases} \tag{25}$$

Equation (25) is almost pedantic in keeping all indexes and variables explicit. It is important to realize that each line corresponds to a different injection level (and then a different value for $I$, $V$, $V_J$), and that the whole system holds for a same frequency $\nu$.

This means that a couple of spectra is sufficient to solve for $L\alpha_T(\nu)$ and $Lg_m(\nu)$ and then to reconstruct $g(\nu)_i$, for all spectra $i$, by Equation (23) or Equation (24). Here we assume that even the end user of commercial devices may access the value of $L$ by a single microscopic image of a laser diode out of the purchased lot. For edge emitters, $L$ is simply the physical extension of the chip along the direction of the optical cavity. This is not the case for vertical emitters, as VCSELs, where the length of the optically active path is different from the length of the resonating cavity.

### 4.1. A Step by Step Example for the Extended Hakki–Paoli Method

Let us apply the method, step by step, to the same device considered in Figure 3.

We start in Figure 8a with six experimental spectra, measured at different currents (7, 8, 9, 10, 11, and 12 mA) on a device (the same as in Figure 3) whose threshold current was $I_{th}$ = 13 mA. We then apply (Figure 8b) the Hakki–Paoli method to calculate the ratio $|F|^2_{MAX}/|F|^2_{min}$ of the envelopes of maxima and minima of the FP resonances, whereas the

enhanced peak due to the DFB grating has been computed separately. In Figure 8c we have the resulting S curves as given by Equation (23).

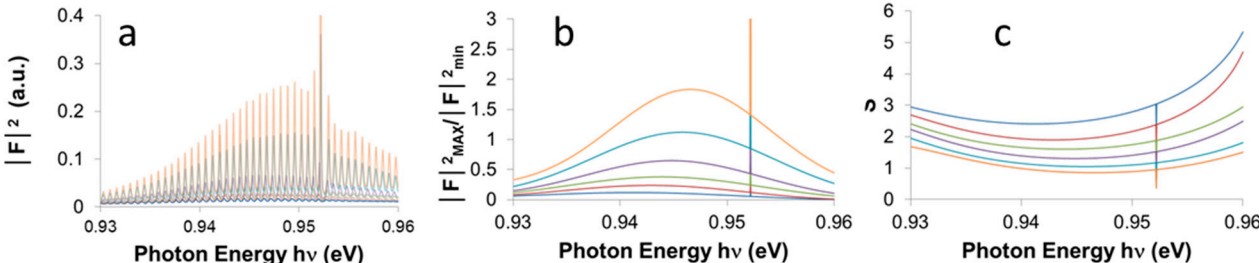

**Figure 8.** (**a**) Six spectra acquired from the same device as in Figure 3 in the sub-threshold region at different currents; (**b**) the ratios between maxima and minima for each spectrum; (**c**) the corresponding S functions (Equation (23)) according to the Hakki–Paoli method.

We then calculate, for each spectrum, the corresponding value of $V_J$ (Figure 9a) by means of Equation (19) and after having calculated the value of $R_S$ (7.2 Ω for the given device). This allows us to plot (Figure 9b) the H curves (Equations (19) and (21)).

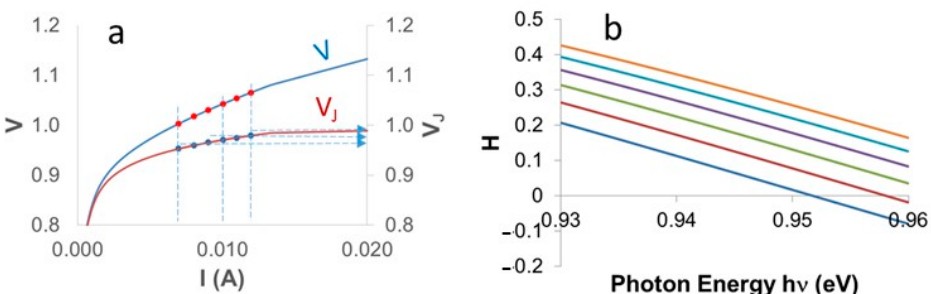

**Figure 9.** (**a**) Calculation of the values of $V_J$ for each spectrum in Figure 8a; (**b**) the corresponding H functions according to Equations (21) and (24).

We now have a set of spectral values for the *S* and the *H* functions for each spectrum *i*. This means that, for each frequency in the spectrum, we have a system of *n* linear equations, with *n* the number of acquired spectra. For each equation, *S* and *H* are known, whereas $L\alpha_T$ and $Lg_m$, which are assumed independent on the injection level, are unknown.

The linearity of Equation (25) with respect to the unknowns $L\alpha_T$ and $Lg_m$ makes the simple least square method well suitable for best fitting experimental data.

Figure 10 displays the six gain curves, one for each spectrum, and the common $g_m$, $-g_m$ boundaries of the general gain function, as well as the spectral loss function $\alpha_T$. The value of *L* resulted, after a simple inspection by scanning electron microscopy, to be $L = 258 \ \mu m$.

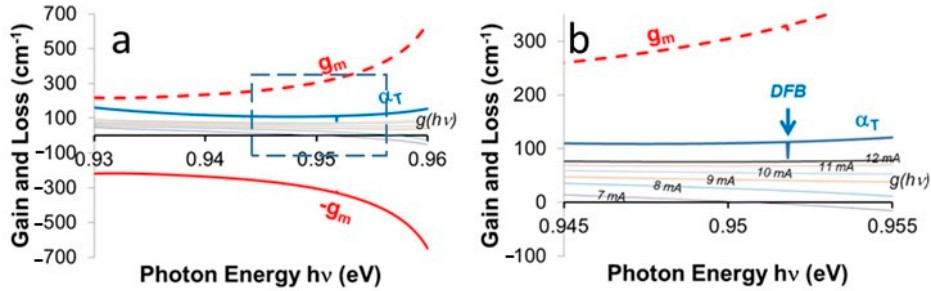

**Figure 10.** (**a**) Gain, loss and absorption spectra for the device under test; (**b**) detail of the boxed area in (**a**).

The detail labelled DFB in Figure 10b is the singular point corresponding to the energy of photons diffracted by the DFB grating. It is interesting that it affects only the loss function $\alpha_T$. The experimentally calculated absorption curve $g_m$ and each gain spectrum $g$ are not modified by the diffracting structure. This is in agreement with the theoretical expectations: the DFB grating forces photons of a selected wavelength to stay inside the optical cavity longer, and then to have a reduced loss rate, than any other photon in the emission spectrum.

The other relevant detail in Figure 10 is that none of the gain curves is anywhere higher than the minimum of $\alpha_T$.

This agrees with the results in Equations (5) and (12), where we saw that gain never can exceed total losses, but tells something more: once one single line in the spectrum first reaches the condition $g(h\nu) = \alpha_T(h\nu)$, it forces $V_J$ to clamp, which in turn blocks gain at any other frequency, no matter if that frequency did not reach the perfect balance between gain and losses.

This last statement has clear experimental evidence when we plot $g(V)$. Let us first consider Figure 11a. It plots gain at the frequency of the DFB peak as a function of both the internal voltage $V_J$ and the external voltage $V$.

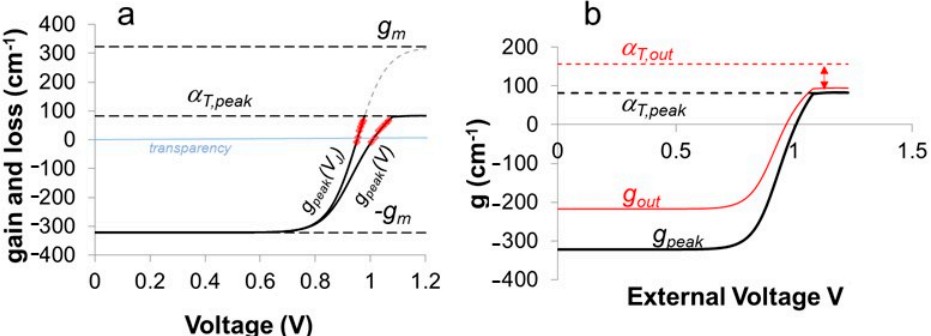

**Figure 11.** (**a**) Gain at the DFB peak as a function of both the internal voltage $V_J$ and the external voltage V; (**b**) gain and loss measured at the DFB peak and at a different emission line (suffix "out"), as functions of the external voltage V.

We can appreciate that the two curves stop at a maximum value because in Equation (21) we used that same measured $V_J$ that saturates itself, as plotted in Figure 6. Moving to Figure 11b, we now plot gain at the spectral peak, as before, and also at a different spectral line, out of the peak, together with the respective calculated value for the loss coefficient $\alpha_T$: only gain at the DFB peak emission saturates exactly at $g = \alpha_T$. All other lines do saturate as well, but stay lower than the corresponding loss term.

The red dots aligned along the gain curves in Figure 11a correspond to those same dots that in Figure 9a indicate the $I,V$ pairs corresponding to each of the six spectra acquired for gain measurements at six different currents. It is noticeable that all of them belong to the sub-threshold regime. In other words, gain never did reach $\alpha_T$ in any spectrum, and nevertheless it has been possible to predict it. On the contrary, the internal voltage $V_J$ spanned the whole sub-threshold and above-threshold ranges, and, when introduced in Equation (21), caused the saturation of $g/g_m$ at exactly the predicted value for the sole peak emission line.

We can graphically summarize what experiments showed by suitably modifying the surface plot of Figure 7. Now (Figure 12) we introduce a surface $\alpha_T$ that intersects the $g$ surface.

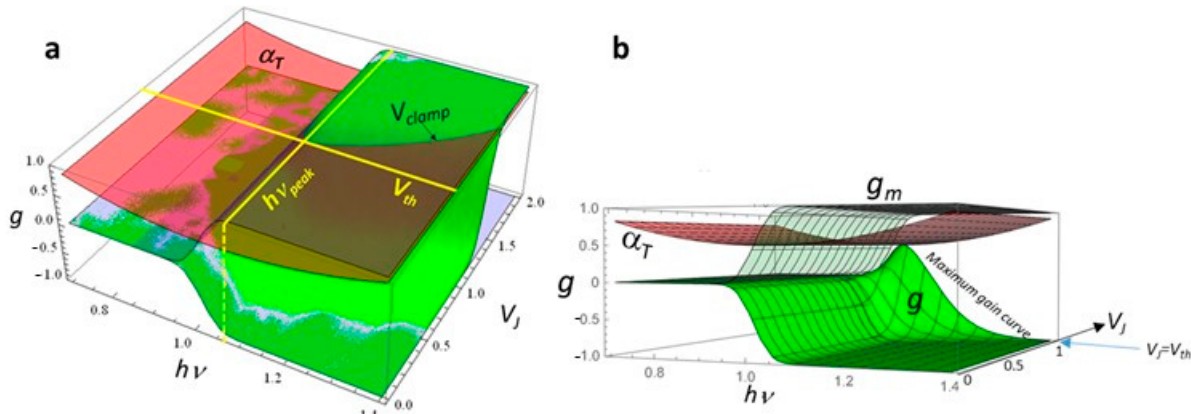

**Figure 12.** (**a**) Plot of the intersection of the gain g and loss $\alpha_T$ surfaces; (**b**) the same plot, limited to the $V_J < V_{th}$ range.

The intersection curve represents the value $V_{clamp}$ of the internal voltage $V_J$ that would be necessary to lead a specific frequency to reach the laser threshold. We know that, as far as one spectrum line will reach its threshold, it will cause the clamp of the quasi-Fermi levels, and then gain g will clamp as well across the whole spectrum.

That curve $V_{clamp}(h\nu)$ has then a minimum along the $V_J$ axis, whose coordinates define both the (internal) threshold voltage $V_{th}$ and the emission peak $h\nu_{peak}$.

The mathematical representation of $V_{clamp}$ follows the inversion of Equation (21), after setting $g = \alpha_T$:

$$qV_{clamp}(h\nu) = h\nu + 2kT \ln\left(\frac{g_m + \alpha_T}{g_m - \alpha_T}\right) \tag{26}$$

$$qV_{th} = qV_{clamp}(h\nu)\Big|_{min} \tag{27}$$

The experimental $V_{clamp}$ curve for the set of data that generated Figures 5, 6 and 8–10 is plotted in Figure 13. In addition to the evidence for the peak at the DFB resonance, it is also possible to appreciate which emission line and the corresponding threshold voltage would be the laser peak in case of absence of the DFB grating.

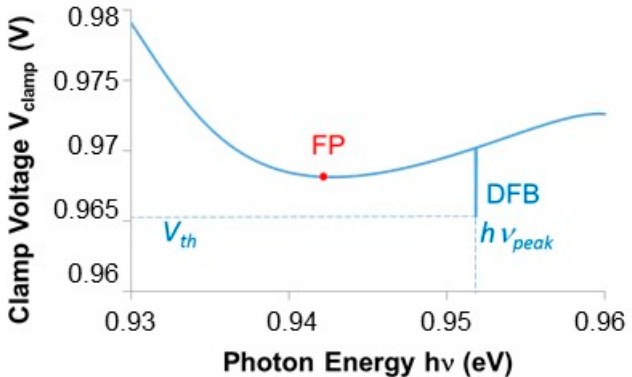

**Figure 13.** Spectral clamp voltage $V_{clamp}$, indicating the threshold voltage $V_{th}$ and the peak emission energy $h\nu_{peak}$. The dot labelled FP indicates the values that $V_{th}$ and $h\nu_{peak}$ would acquire in the case of no DFB grating.

The relationship between gain and voltage has a numerical peculiarity that is worthy of comment. If we plot the function $H$ at a fixed photon energy as a function of the external voltage in a narrow range close to transparency, and also plot the laser current $I$ on a logarithmic axis, we get two nearly parallel straight lines (Figure 14).

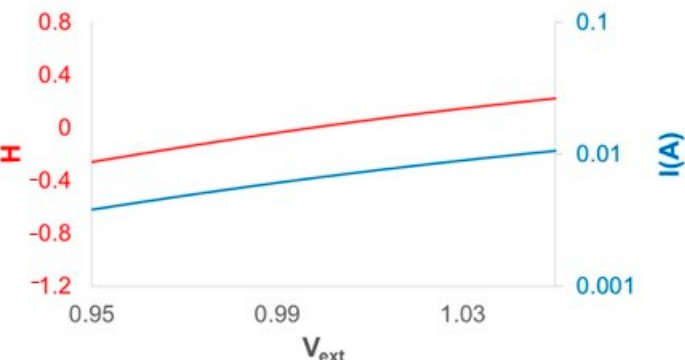

**Figure 14.** *H* function at a fixed frequency (left), close to transparency and laser current in logarithmic scale (right) as functions of the external voltage.

If for the current *I* this means that the series resistance, in the displayed range, is not too large, then the function *H*, despite its mathematical non-linearity, is approximately linear around transparency. If now we recall the red dots in Figure 11a, and realize that they represent the standard conditions for gain measurements, we see from Figure 14 that a $g/g_m \propto \ln(I)$ numerical relationship between gain and current can be not only proposed, but also confirmed by experiments [34]. This is an empirical relationship, that fails when one explores injection conditions far from transparency, where indeed upper and lower gain limits $-g_m$, $g_m$ cannot be associated with the current range 0,∞.

## 5. Non FP-Resonating Cavities

None of Equation (18) works on non-FP resonating cavities (null reflectivity at least on one of the two facets). In such cavities, resonances can be caused by external reflectors, external cavity tunable devices, or by Bragg structures either internal (DFB) or external (DBR). In any case, apart from the solitary spectral peak corresponding to the tuning frequency, most of the spectral emission does not display any modulation caused by resonance.

We can describe those cavities as infinitely lossy, which means $\rho = 0$ in Equation (17) that reduces to $|F|^2 = |F_0|^2$.

Gain measurement then requires a different equation to be coupled with Equation (21). In particular, we need an explicit expression for $|F_0|^2$, that is, the intensity of the non-modulated spectrum.

We can start from Equation (6) and look for an explicit expression of the spontaneous emission rate $R_{sp}$ as a function of gain parameters. The task requires to first recall a classical result in laser diode theory [15], in the comfortable form of [33], and then manipulate it somewhat [66]. We get

$$R_{sp} = \frac{8\pi v^2}{c^2} \frac{g_m}{4} \left( 1 + \frac{g}{g_m} \right)^2 \tag{28}$$

Equations (8) and (9) accordingly become:

$$P_{TOT} = Vol \cdot \int h v \frac{2\pi v^2}{c^2} \left( 1 + \frac{g}{g_m} \right)^2 \frac{g_m \alpha_T}{\alpha_T - g} d v \tag{29}$$

$$I_{ph} = q \cdot Vol \cdot \int \frac{2\pi v^2}{c^2} \left( 1 + \frac{g}{g_m} \right)^2 \frac{g_m \alpha_T}{\alpha_T - g} d v \tag{30}$$

If we now consider an experimental power spectrum, measured at a given current $I$, and made of a sequence of $N$ discrete values $f_n$, $n = 1, 2 \dots N$, we recognize that

$$\sum_{n=1}^{N} f_n \propto P_{TOT}$$
$$q \sum_{n=1}^{N} \frac{f_n}{h\nu_n} \propto I_{ph}$$

(31)

We are close to link experimental spectra, without any resonance, to some measurable quantities, provided we find the proportionality constants. If, on one side, the experimental measurement of $P_{TOT}$ requires special instruments as an integrating sphere, the experimental value of $I_{ph}$ comes straight from the simple analysis of the $I(V)$ characteristics of the laser diode [58].

In short, the current $I$-$I_{th}$ exceeding the threshold $I_{th}$ identifies with $I_{ph}$ (while under threshold $I_{ph}$ is hidden by the non-radiative part $I_{nr}$ of the total current $I$). One then calculates a constant $\chi$ using one or more spectra acquired over the threshold

$$\chi = \frac{I - I_{th}}{q \sum_{n=1}^{N} \frac{f_n}{h\nu_n}}, \qquad I > I_{th}$$

(32)

The $\chi$, calculated for a restricted range of currents reveals the proportionality relationship that holds between $I$-$I_{th}$ and the integrals of the experimental spectra across the whole range of currents.

Now, if we multiply all spectral values in each spectrum, including the sub-threshold range, by $\chi$, we get

$$q\chi \sum_{n=1}^{N} \frac{f_n}{h\nu_n} = I_{ph}, \qquad \forall I$$

(33)

Figure 15 plots three sets of data from a DFB laser diode with non-reflecting facets. The first set is the over-threshold current $I$-$I_{th}$ that we identify with $I_{ph}$ (the logarithmic vertical scale cuts off the sub-threshold current range, for which $I$-$I_{th} < 0$). The second set is the optical emission $P_{OUT}$ as measured by a photodiode, upscaled by a suitable multiplier (whose dimensions are the same as $q/h\nu$ in Equation (31)) up to overlap with $I$-$I_{th}$. In this way, the plot of $P_{OUT}$ transforms into the plot of $I_{ph}$ across the whole sub-threshold and over-threshold ranges. The third set of plotted data is a sequence of dots, corresponding to approximately 30 spectra. Each dot is the sum of all experimental values in the spectrum, multiplied by the factor $\chi$, (Equation (32)), identical for all dots, which brings the whole sequence to align along the curve of the upscaled $P_{OUT}$.

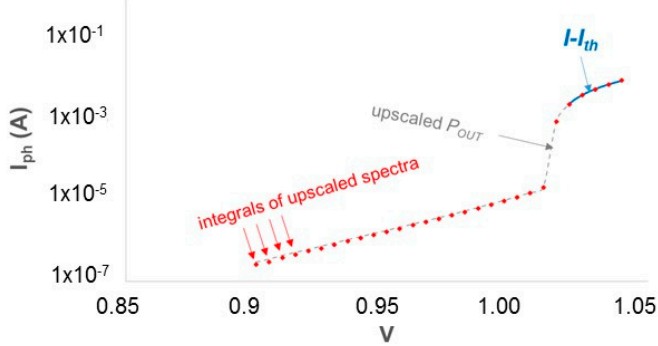

**Figure 15.** Three curves drawing the current $I_{ph}$ as a function of the external laser voltage V. The first is the difference $I$-$I_{th}$ measured from the DC characteristics. The second (upscaled $P_{OUT}$) is in practice the photocurrent created in a monitor diode by the laser radiation, suitably upscaled to coincide with the over-threshold $I$-$I_{th}$. The third is made of the integrals of several spectra, multiplied by a suitable common constant that brings their value to overlap the upscaled $P_{OUT}$ curve.

If now we compare Equations (33) and (30) we get

$$q\chi\frac{f_n}{h\nu_n} = q \cdot Vol \cdot \frac{2\pi\nu_n^2}{c^2}\left(1+\frac{g}{g_m}\right)^2\frac{g_m\alpha_T}{\alpha_T-g}\Delta\nu_n \tag{34}$$

The term $\Delta\nu_n$ replaces the differential element $d\nu$ in integral forms and represents the optical frequency sampling step in numerical data.

Rearranging Equation (34) one can write

$$\frac{\alpha_T-g}{g_m\alpha_T} = \frac{q \cdot Vol \cdot \frac{2\pi\nu_n^2}{c^2}\Delta\nu_n}{q\chi\frac{f_n}{h\nu_n}}\left(1+\frac{g}{g_m}\right)^2 \tag{35}$$

Despite its unfriendly appearance, the right-hand side of Equation (35) is a known quantity at each frequency of each spectrum. Let us shortly indicate the following spectral function $h_i$ where the suffix $i$ identifies the spectrum (a dot in Figure 15), and the index $n$ simply recalls that experimental spectra are made of discrete values for the optical frequency $\nu$ and the photon energy $h\nu$:

$$h_i = \frac{q \cdot Vol \cdot \frac{2\pi\nu_n^2}{c^2}\Delta\nu_n}{q\chi\frac{f_n}{h\nu_n}} \tag{36}$$

Recalling then the definition of the function $H$, Equation (24), we obtain

$$\frac{1}{g_m}-\frac{H_i}{\alpha_T} = h_i(1+H_i)^2 \tag{37}$$

This is the relationship that replaces Equation (25). All terms in Equation (37) are spectral. At a fixed frequency, $h_i$ and $H_i$ are different from spectrum to spectrum, whereas $g_m$ and $\alpha_T$ are the same for all spectra. As for the resonating cavities, two or more spectra allow us to solve Equation (37) and get $g_m$ and $\alpha_T$. Equation (24) then reconstructs $g$.

Figure 16 reports the results for the laser diode of Figure 15, removing the DFB contribution.

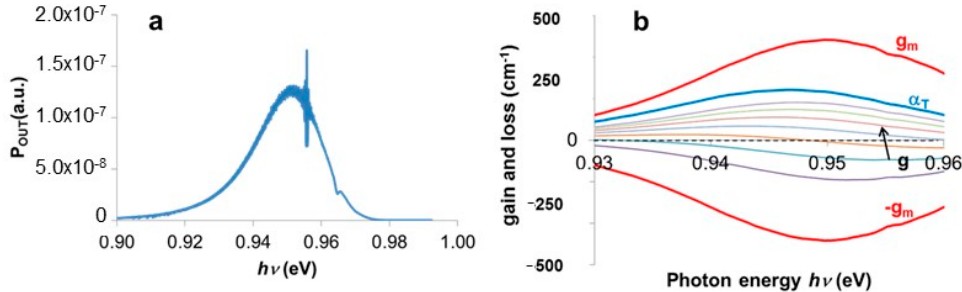

**Figure 16.** (**a**) One of the spectra of the DFB laser with a non-resonating cavity; (**b**) gain curves for the non-reflecting cavity.

The most challenging step in this case of non-resonating cavities is the need to measure the volume *Vol* of the active region that enters calculation through Equations (29) and (30). This is much more demanding than measuring the sole length $L$ of the optical cavity, because the thickness of the active layer (or the cumulative thickness in the case of multi quantum wells (MQW)) is beyond the resolution of the scanning electron microscope (SEM): determination of the thickness of the active region requires a transmission electron microscope (TEM). As for the previous methods, it requires the sacrifice of one single device.

Figure 17 shows the same structure employed in Figure 1c to illustrate graphically the mean free path for photons inside the optical cavity. It now has the measured dimensions,

and shows the number and thickness of multi quantum wells that build up the active region, whose length *L* and width *W* result from SEM inspection. It is interesting to observe that for DFB structures, the physical value of the pitch *a* leads to the measurement of the refractive index by means of its ratio with the measured peak wavelength.

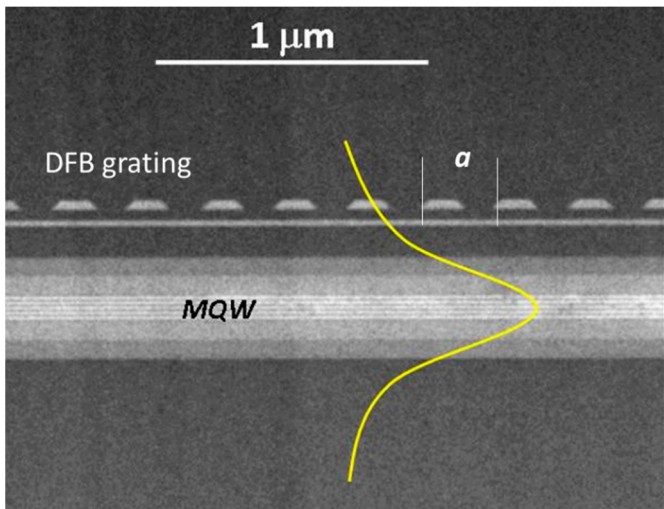

**Figure 17.** Longitudinal cross-sectional TEM image of the active layer and the surrounding structure in a DFB laser diode. The yellow curve indicates graphically the expected distribution of the optical intensity.

## 6. Gain and Reliability

The previous sections aimed to extract gain data from experimental measurements, extending and improving existing procedures. The Introduction linked such measurements with reliability. This section will then develop that relationship, which is clear within the community of reliability engineers, but may be worth some recapitulation for a wider audience of readers.

Reliability mimics several concepts from human life, which helps at an intuitive level, but sometimes is misleading. In particular, the concept of *failure*, usually associated with death, is instead closer to that of *attitude to work*. A football player, a singer, a pilot can close their activity and have still a long life to enjoy. The definition itself of reliability is the "Attitude (probability) of a device to perform assigned operation(s) for a given time under defined operating conditions" [35]. Such a definition calls for the definition of a *failure criterion* that can be different from device to device and even for identical devices differently employed. For instance, the same detuning $\Delta\nu$ in the peak frequency of a single-mode laser diode can flag a failure in WDM applications and yet keep perfect operativity in optical pumping.

The example also reveals that many different parameters can cause a faulty state: in the list of specifications of a commercial device, any element that first drifts outside the accepted tolerances will define the time-to-failure (TTF) of that device. Even in the same lot of identical devices operating in identical conditions it may happen that the fault condition occurs because of different failure criteria in different elements of the lot.

Because the mean-time-to-failure (MTTF) is the ultimate prediction sought by reliability testing, one can guess the hardness and complexity of the task.

Even more, if one wonders about the physical origin of a failure, the concepts of *failure modes* and *failure mechanisms* enter into play: mechanisms are the physical causes and modes the observable effects of a degradation phenomenon, and there is not a one-to-one match. There is a perfect analogy with diseases and symptoms in living beings: a fever can be caused by several pathologies. It should be clear that failure criteria, and then TTF and MTTF, rely on failure *modes*.

Medical care, or even health statistics, should look to the root causes, and not to the fever, to be effective. In solid state devices, faulty elements are never treated for recovery. Nevertheless, the discovery of the failure mechanisms underlying the observed failure modes is the key for any corrective action (a duty of the device manufacturer) or for the design of suitable screening of the incoming lots (a task of the customer).

Let us transfer the above concepts to our case.

For laser diodes and light emitters in general, it is common and natural to consider the reduction of emitted optical power below some specific level as the main failure criterion. A plot as in Figure 1a or Figure 2a (the so-called *LI curves*) is the maximum information usually provided in the datasheets accompanying a commercial device. It essentially displays only two quantities: the threshold current $I_{th}$, corresponding to the abscissa of the kink at turn-on, and the total efficiency $\eta$ of the device, given by the slope of the curve for currents $I > I_{th}$. The operating point then corresponds to a specific $(I, P_{OUT})$ pair, and a power reduction of, say, 20% can represent the failure criterion.

Now, let us focus on Figure 18, from reference [69]. It is just an example of the variety of ways that a *LI* curve can change upon the activation of a degradation mechanism. It is not important here to give details of that case, which was a controlled experiment on the transitory effects of mid-energy proton irradiation on the performances of two different commercial lasers. What is relevant for the sake of this paper is the evolution of the curves, starting from their initial state (the dashed lines). We appreciate both a correlation between an increase in $I_{th}$ and the decrease in $\eta$, but also a sequence of curves where only the threshold current increases, and the slope remains the same.

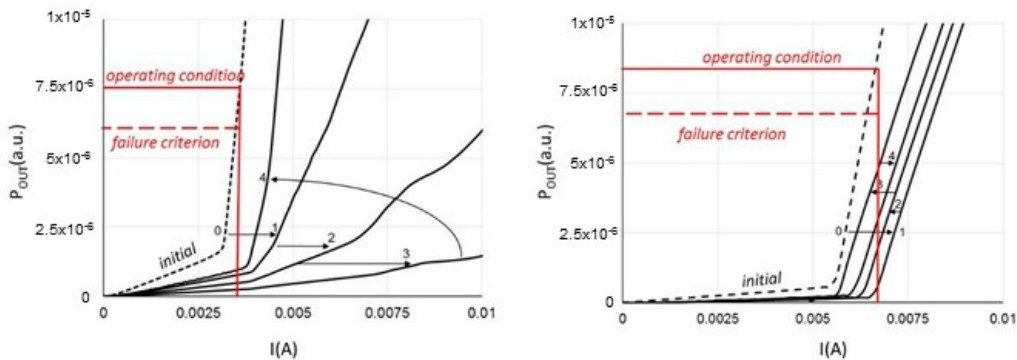

**Figure 18.** The LI curves of two different commercial laser diodes after proton irradiation [69].

The variations in $I_{th}$, and $\eta$ are the *failure modes*. In this case, the failure mechanism was known: proton irradiation. Any explanation of the observed phenomena requires one to link proton irradiation with the physics and technology of the specific devices. In particular, we must assume that protons create enhanced recombination of electron and holes, and also optical capture centers for photons, which means an increase in the internal losses $\alpha_\iota$.

Now, the threshold current may be expressed with excellent approximation [67] by

$$I_{th} \approx I_{th0} \exp\left(\frac{\alpha_T}{g_m/4}\right) \tag{38}$$

Here $I_{th0}$ is the ideal threshold current at zero losses. Appendix A recalls that $I_{th0}$ corresponds to the value of the non-radiative component $I_{nr}$ of the total current $I$ flowing across the device at the threshold voltage $V_{th}$, Equations (26) and (27). It depends on the amount of electron-hole recombination, but does not depend on optical losses. In contrast, total losses $\alpha_T$ exponentially influence the threshold current in conjunction with the gain limit $g_m$.

Efficiency $\eta$, in turn, depends on the ratio $\frac{\alpha_m}{\alpha_T}$

$$\eta = \frac{h\nu}{q}\eta_C\frac{\alpha_m}{\alpha_T} \tag{39}$$

with $\eta_C$ the coupling efficiency of the light collection apparatus [34]. It does not depend on gain or electron-hole recombination. For the given case, the hypothesis of diffusing protons across the two different 3D geometries of the devices allowed us to reconstruct the link between the time varying proton density and the observed variations in the *LI* curves, up to measurement of a diffusion coefficient corresponding to the known value for interstitial hydrogen diffusing in gallium arsenide and indium phosphide at room temperature.

For devices that failed during operating life or during accelerated life tests, the leading failure mechanism is not known a priori. The observed failure mode can be any of the cases in Figure 18, or even something different (increasing threshold current *and* increasing efficiency, or changing slope with constant $I_{th}$). The role of the methods proposed within this paper is to allow the separate measurement of gain $g$, of its boundaries $\pm g_m$, and of the total loss coefficient $\alpha_T$. Recalling that gain depends on the properties of the active material, Equation (21), and not on the size and shape and boundary reflectivity of the optical cavity, whereas losses change because of defects inside or at the edges of that cavity, the separation of gain and loss contribution addresses diagnostics towards material and cavity.

## 7. Discussion and Conclusions

Several points in the paper require some discussion because of their apparent disagreement with the standard approaches to laser diode theory.

In Section 2, the introduction of the continuity equation for photons may look weakly founded and only based on a heuristic comparison with a similar equation for minority carrier in semiconductors. In particular, the "diffusion coefficient" seems merely built by analogy. On the contrary, the proposed approach relies on the construction and solution of a rate equation for the uniform domain, anchored to the balance among electron, hole, and photon densities and on their interaction probability according with the Einstein treatment of the black body radiation. Ref. [67] summarizes in detail several papers that the author published throughout several years on that subject. The results are in perfect agreement with the foundation papers, in particular with respect to the formulation of the photon density for the uniform domain (Equation (5)) and then, necessarily, for the related quantities of gain and emitted optical power. Section 2, then, essentially points out that optical emission at low current does not involve the whole optical cavity. Figure 1 summarizes that result.

Moving to gain measurements, all of the proposed methods have a common point: the separate measurement of gain and total losses. All of them work with spectral data in the sub-threshold range to measure gain and losses. However, gain itself saturates at threshold, which makes spectra well above threshold simply useless. This helps because we may neglect the non-idealities observed at high injection, pointed out, for instance, by the range of high currents for which the LI curve deviates from linearity. For the same reason, also the calculation of the series resistance $R_S$, so important for our method, can refer to the $I(V)$ range close to threshold. In practice, the best value of $R_S$ is that one that, starting from the experimental $I$ and $V$, gives back the largest set of current values for which $V_J$ remains constant.

In example, Figure 19 plots the laser current $I$ and the photocurrent current $I_M$ measured by a monitor diode (which is proportional to the emitted optical power $P_{OUT}$), as functions of the internal voltage $V_J$. The bold lines for both curves represent the range of validity of the ideality approximation. The thin dashed lines show the range where $R_S$ looks to change, as indicated in the note after Figure 5 [68].

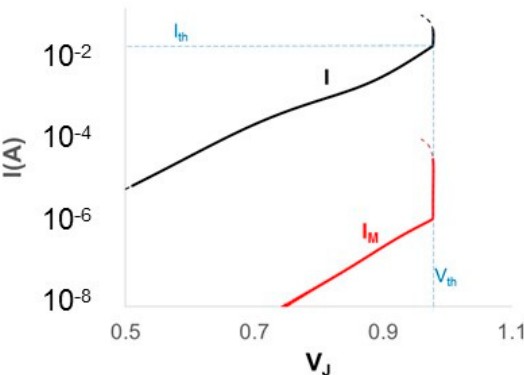

**Figure 19.** Laser current I and monitor current $I_M$ as functions of the internal voltage $V_J$ for the laser of Section 4.

Data in Figure 19 have been acquired by driving the laser diode with a linear voltage ramp from 0 to 1.5 V at steps of 2 mV. By measuring $V_J$ with an accuracy of 1 mV, the value $R_S = 7.75\ \Omega$ generated a set of 83 voltage values fixed at the threshold voltage $V_{th} = 0.978$ V. Any other value of $R_S$ reduced the number of elements in that set. As an aside, the measured value of $V_{th}$ gives an upper estimate $E_g \approx 0.978$ eV of the bandgap of the active material, as indicated by Equations (26) and (27), which show that $qV_{th}$ is somewhat larger than the minimum available photon energy $h\nu$. This is obviously consistent with the requirement $\phi_n - \phi_p > E_g$, which is the fundamental rule for population inversion in semiconductors.

Section 3 is a summary of the gain measurement methods based on the analysis of spectral data modulated by FP and DFB/DBR resonances. That section summarizes the current state of the art for gain measurements from spectral analysis. It also points out the two difficulties of the Hakki–Paoli and the derived methods: a) the measurement not of gain g but of the difference $g$-$\alpha_T$ and b) the impossibility to measure gain from non-resonating cavities.

Moreover, a check on the linearity of the spectral response (or even on the perturbation introduced by some ambient light) becomes available from the reconstruction of maxima and minima $|F|^2_{MAX}, |F|^2_{\min}$, as in Figure 3, starting from the $|F|^2_+, |F|^2_-$ functions, calculated by the moving averages methods. If the calculated curves do not run along the top and the bottom of a spectrum, then that spectrum does *not* follow the ideal Equation (17) [65].

The key step is Section 4, where Equation (21), recovered from the theoretical foundation of laser diode physics, decouples gain $g$ from losses $\alpha_T$ and introduces the upper and lower gain limits $\pm g_m$ as measurable quantities as well.

Last, the final section, Section 5, proposes a way for extending gain measurements to non-resonating cavities. It requires the calculation of the single-trip term of the photon intensity, that is the term $|F_0|^2$ that appears in all lines in Equation (18), and that is duly removed by the evaluation of ratios.

Here the critical point is the accurate quantification of a multiplier that includes the volume *Vol* of the active region, which is not a simple piece of information to get. In ref. [66] it is shown that gain curves, calculated with the Hakki–Paoli, the Cassidy, or the Vanzi methods, and the method of Section 5 for the same set of data are consistent.

A byproduct of this last section refers to Figure 15. It is a check of the consistency of the measured spectral intensities. If it is impossible to find a multiplier common to all spectra that aligns their integrals (dots) along the plot of the monitor output, then one must check if the spectrometer has some setup that adjusts gain accordingly with the spectrum intensity.

The general applicability of the proposed (or recalled) methods stands in Equation (21) and in its demonstrated measurability. The spectral function $g/g_m$ is a universal function of the junction voltage $V_J$. This means that, once calculated $g_m$, gain can be calculated at any injection level. It should be kept in mind, in any case, that gain stops at threshold. The other

models [34,59] (linear, logarithmic, with or without bias) all work close to transparency, that is close to threshold, and fail in predicting gain at very low and very high currents.

The overall relevance of the paper, in the author's belief, is for reliability studies. During life tests, or after degradation after some operating life, alterations occur inside the optical cavity that change the initial state of the device. Losses become a monitor of many degradation kinetics, and then it becomes crucial to measure them and not to assume their values from databases. However, reliability investigation is mandatory for the end user of any kind of devices, at both the beginning and the end of their operating life, that is, for qualification of the incoming purchased lots and for diagnostics of failures from field operation. Here we close the loop, with a reference to the beginning of this paper, that is its title: optical gain in commercial laser diodes might be a powerful tool for characterization and reliability of such devices, but must break free from the need of external information or excessively demanding procedures and instruments. The paper proposes a general approach, based on the standard measurement of a number of spectra and of the DC electrical characteristics, that aims to ease the task.

The paper continuously took care to refer even its most unusual results and interpretations to the very foundation of the laser diode history. In that spirit, we will then conclude this paper by also referring to the historical link between gain and reliability. Indeed, the original paper from Hakki–Paoli [40] that for more than half a century drives the most part of gain measurements, is titled: "*cw degradation at 300 °K of GaAs double-heterostructure junction lasers*". *II. Electronic gain*".

**Funding:** This research received no external funding.

**Acknowledgments:** The author is deeply indebted with D.T. Cassidy for his authorative and kind encouragement during the development of this paper.

**Conflicts of Interest:** The author declares no conflict of interest.

## Appendix A. Current and Voltage in a Laser Diode

The physically sound relationship between optical gain and injection level in a laser diode is given by the Lasher and Stern formula [15] that here appears in the form of Equation (21) after replacing the quasi-Fermi levels with the internal voltage $V_J$, Equation (19), proposed by Barnes and Paoli [39]. The simplifying approximation of Equation (22) has been proposed and used by the author in all his papers [55–69] for the sake of referring elusive quantities as the electron and hole energies and densities to the measurable alternative quantities $h\nu$ and $qV_J$. The result is that Equation (21) becomes a supplementary independent expression for optical gain that, coupled with any of the other measurement methods resumed in Sections 3 and 5, leads to the separate measurement of gain and losses.

It is not usual to refer the performances of a laser diode to the applied voltage $V$; it is the forward current $I$ that instead plays that role: spectra in Sections 3–5 have been identified by the corresponding current, and $LI$ curves as in Figures 1a, 2a and 18 have the current $I$ on their abscissas, and the laser threshold is identified by the threshold current $I_{th}$. It seems then advisable to develop an alternative formulation of the proposed methods based on the current $I$ instead of the voltage $V$ by means of a suitable $I(V)$ relationship for a laser diode. This has been one of the main tasks of the laser model developed by the author in previous years in the reference papers for this appendix.

The kernel is here summarized:

All currents in an ideal laser diode come from recombination of electrons and holes. Part of such recombinations cause light emission and the remaining part does not. The two contributions to the total current $I$ have been named, respectively, $I_{ph}$ and $I_{nr}$. The two currents share the same junction bias $V_J$, and they are additive, as for two circuital elements connected in parallel.

$$I(V_J) = I_{ph}(V_J) + I_{nr}(V_J) \tag{A1}$$

The non-radiative component $I_{nr}$ does not differ from the standard Shockley current

$$I_{nr}(V_J) = I_{nr0} \left[ \exp\left( \frac{qV_J}{kT} \right) - 1 \right] \tag{A2}$$

where $I_{nr0}$ is the saturation current.

The radiative current $I_{ph}$ is the kernel of the laser: it corresponds to an ideal device where all recombination occurs inside the optically active region, is spatially uniform, and is completely converted into light. Would that device exist, its current-voltage characteristics should be:

$$I_{ph} = I_{ph0} \frac{\alpha_T + g_m}{\alpha_T \left[ 1 + \exp\left( \frac{qV_J - h\nu_{peak}}{2kT} \right) \right]^2 + g_m \left[ 1 - \exp\left( \frac{qV_J - h\nu_{peak}}{kT} \right) \right]} \left[ \exp\left( \frac{qV_J}{kT} \right) - 1 \right] \tag{A3}$$

where $h\nu_{peak}$ is the photon energy at the spectral peak. That peak photon energy is that value of $h\nu$ that leads the denominator in Equation (A3) to vanish for the minimum value of $qV_J$, which corresponds to the condition for voltage clamp introduced in Equations (26) and (27).

The peak photon energy is larger than the bandgap $E_g$ of the active material, so that for any bias $qV_J < E_g$ the exponentials in the denominator are negligible with respect to unity, and the whole fraction approaches the unity. In that range $I_{ph}$ behaves as a standard Shockley current, with saturation current $I_{ph0}$. On the contrary, when the bias $qV_J$ approaches $h\nu_{peak}$, the current $I_{ph}$ increases without limits.

In a real laser diode, $I_{ph0}$ is much less than $I_{nr0}$, so that below the threshold $I_{nr}$ dominates over $I_{ph}$. Figure A1 plots the three currents of Equation (A1).

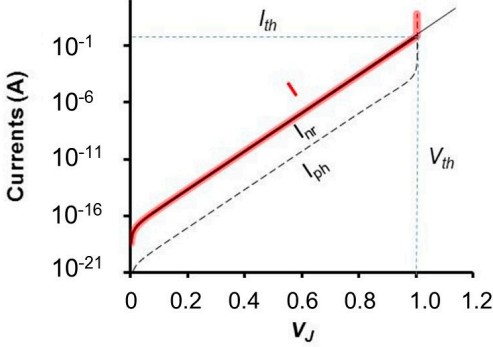

**Figure A1.** The total current $I$ (red bold line) and its components: the radiative current $I_{ph}$ (dashed line) and the non-radiative current $I_{nr}$ (thin solid line). The threshold voltage $V_{th}$ and current $I_{th}$ are also indicated as the coordinates of the sharp transition in $I$.

In a real device, extra currents flow at low bias, mostly due to the lateral sides of the active region, and become negligible as bias approaches threshold. Additionally, a series resistance, typically of a few ohms, plays a role, masking the ideal vertical transition shown in Figure A1 and reconstructed from experimental data in Figure 19.

In any case, the threshold current $I_{th}$ results as the value of the non-radiative current $I_{nr}$ at the clamp value of $V_J$. This conclusion agrees with the considerations in Chapter 2 of ref. [34].

Substituting Equation (21) in Equation (A2), and neglecting the unity because we are certainly considering deep positive bias, we get

$$I_{th} = I_{nr}(V_{th}) = I_{th0} \left[ \frac{1 + \frac{\alpha_T}{g_m}}{1 - \frac{\alpha_T}{g_m}} \right]^2 \tag{A4}$$

$$\text{with } I_{th0} = I_{nr0} \exp\left(\frac{h\nu_{peak}}{kT}\right) \tag{A5}$$

The quantity $I_{th0}$ indicates the theoretical lower limit for $I_{th}$ in the ideal case of zero losses. Alternatively, it represents the value of the non-radiative current when $qV_J = h\nu_{peak}$, which corresponds (see Equation (21)) to $g = 0$. It is the *transparency* condition, that is close to the threshold condition stated by Equations (26) and (27).

We see an explicit link between the measurable $I_{th}$, the gain extremal $g_m$, and the total losses $\alpha_T$. It is a particular case of a more general relationship among current, gain, and losses that, based on Equations (A1)–(A3), (21) and (22):

$$I = I_{th0}\left[\frac{1 + \frac{g}{g_m}}{1 - \frac{g}{g_m}}\right]^2 + I_0 \frac{\left[1 + \frac{g}{g_m}\right]^2}{1 - \frac{g}{\alpha_T}} \tag{A6}$$

where $I_0$ is the value of $I_{ph}$ (Equation (A3)) at transparency ($qV_J = h\nu_{peak}$).

Being $I_{ph0} \ll I_{nr0}$, as stated before, we have also: $I_0 \ll I_{th0}$. The second term on the right-hand side of Equation (A6) remains completely negligible if $\alpha_T > g_m$, and the gain-current relationship reduces to

$$I = I_{th0}\left[\frac{1 + \frac{g}{g_m}}{1 - \frac{g}{g_m}}\right]^2, \quad \alpha_T > g_m \tag{A7}$$

We can see in Equation (A7) the optical gain in a light emitting diode, whose current $I$ can increase without limits, but gain never exceeds the range $-g_m, +g_m$. No laser threshold will ever be reached.

On the contrary, if $\alpha_T < g_m$ the first term on the right side of Equation (A6) continues dominating over the second one, until $g = \alpha_T$. Approaching that point, current rises to very high values due to the stimulated emission. In practice, the first term stops at $g = \alpha_T$, which corresponds to the threshold condition in Equation (A4).

Both Equations (A4) and (A6) conflict with the most popular empirical equations for the gain-current relationship. It is noticeable that when we consider the power expansion of Equation (A7) close to transparency ($g = 0$), we see that it mimics, up to the fourth term, the power expansion of the function

$$I = I_{th0} \exp\left(\frac{g}{g_m/4}\right) \tag{A8}$$

In the same way, the power expansion of Equation (A4) emulates, for small values of the loss coefficient, the equation

$$I_{th} = I_{th0} \exp\left(\frac{\alpha_T}{g_m/4}\right) \tag{A9}$$

Equation (A8) explains why experimental measurements suggested a logarithmic relationship for $g(I)$, whereas Equation (A9) coincides with Equation (38) and justifies it under the numerical point of view.

In conclusion of this appendix, one could assume that referring gain to voltage or current is almost equivalent. It is not so.

The crucial point is the suitability of an ideal Shockley relationship, as in Equation (A2) and in the sub-threshold behavior of Equation (A3), to represent the $I(V)$ characteristics of a real device. In particular, experiments almost never show an ideality factor equal to unity. In other words, the measured currents in the subthreshold range seem to follow the reduced bias $V_J/N$, with $N > 1$.

The explanation is much more complex than a first glance interpretation: voltage drops in part outside the active region, and then the effective $V_J$ is a fraction of the applied value. This would simply cause the stretching of the abscissa in theoretical and experimental plots

as in, respectively, Figures 19 and A1. The experimental evidence adds one disrupting element: stretching that works well for the sub-threshold regime does *not* apply to the threshold voltage $V_{th}$, as pointed out, together with a possible explanation, by the author in [62]. It is that complexity of the $I(V)$ characteristics that blurs the clarity of the gain-voltage relationship when one attempts to translate it into a functional dependence on the current $I$.

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
