# Peer review of "Optical Gain in Commercial Laser Diodes"

_photonics, doi:10.3390/photonics8120542_

Round 1

Reviewer 1 Report

The paper of Massimo Vanzi is a well written article about gain in laser diodes. In my opinion the paper is some kind of handbook chapter related with laser diodes.

He is an expert in the field of semiconductor laser diodes and gain characterization method what is confirmed in mentioned paper where well known modifications of Hakki-Paoli methods are discussed (H-P, Cassidy and Vanzi). In general I do not have any critical remarks, only some sugestions about the citations - for example line 69 I wolul recommend to use [39-40] instead of [39]-[40] and I woluld recommend to use space between the sentence and citation i.e. text [x] instead of text[x].

Author Response

I thank the reviewer for his/her appreciation of my new paper and the evident knowledge of my previous ones.

Reviewer 2 Report

This is a nice paper about a very relevant topic as the optical gain in laser diodes.

This topic has been the object of many studies, well documented in this paper. Here, the different approaches are reviewed, and an approach to the study of gain and losses in commercial lasers by optical and electrical analysis of the laser properties, aiming to apply it to the analysis of laser degradation.

The approach is based is based on a continuity equation for photons taken as particles. The solution of this equation provides an analogous to the minority carrier diffusion length of carrier transport. This approach can be usable for non resonant cavities, but it must not work for resonant cavities, as the interferences are missed.

The Hakki-Paoli and Cassidy approaches are presented, and compared to the author approach, which will provide  more simplicity.

   Fig. 11a : in the label of the x-axis, please indicate internal voltage or Vj (V)

I don’t see the interest of using c, as eq. 28 is I-Ith

Eq. 29 depends on the frequency interval taken, please comment about the optimum interval

Ref. 41: JAP 51, 3042 (1980)

Author Response

Thank you! 

Your comment about the continuity equation is correct: it does not aim to deal with resonating cavities, but "only" to point out the link between gain, loss and opacity. It is also a sort of preparation of chapter 5, avoiding the cumbersome reconstruction of the balance equation starting from the Einstein formulation of the Black Body radiation, given in ref.70, that is one of my chapters in a new book. I got a suggestion from prof. Dan Cassidy about the possibility of harmonizing wave-particle representations for gain, but it is at the level of a dream.

About fig.11 a, I left intentionally V in the abscissa, because the two curves are one plotted vs. VJ and the other vs. Vext. I hope it can be clear.

About eq.28 the poit is that Iph=I-Ith only for I>Ith. Here I propose to measure Iph BELOW threshold. That is the reason for that ...mess!

Eq.29. You are right. I included the statement about the sampling step just after the equation

Reference 41 (now 42) adjusted.

Thank you again!

Reviewer 3 Report

The submitted manuscript investigates into the relationship between the gain parameter and reliability of laser diodes. My comments thereupon follow:

  1. The fundamental idea of the work is interesting and attractive. When purchasing a new laser diode, it would be highly desirable to know its projected life time. Correlation of this parameter to the active medium gain of the laser diode seems to represent a promising approach to estimation of the laser diode life time. The overwhelming majority of the readers would be interested in practical recommendations as to how one could predict the life time of a particular laser diode. Therefore, such practical recommendations should be provided. What and how to measure in a new laser diode to find out whether or not it will work for a long time? This should be added to the manuscript.
  2. It would be desirable to understand limitations of the proposed method. Is it applicable to any laser diode or this depends upon the type of the gain medium, radiation intensity, wavelength, &c? The field of applicability of the presented results needs to be specified.
  3. On the whole, the text produces a weird impression. One naturally expects to find some dependencies of laser diode life times (how many thousands of hours) upon certain generation parameters of those diodes, but the manuscript gives none. These dependencies should be provided (even if only theoretical), otherwise the work results on reliability of laser diodes are difficult, if not impossible, to apply to real-world devices.

The submitted manuscript may be published in Photonics on provided that the raised issues are addressed in a further revision.

Author Response

Thank you. It has been important that you clarified the possible difficulty of the readers in connecting this paper with Reliability.

I have added a new chapter,  number 6, trying to explicitly show how gain measurements can address diagnostics. 

I even attempted to write an appendix summarizing the key points in Physical Reliability, and mostly the many misleading interpretations that do survive. It was too huge in the short time allowed for revision. Linking gain to a lifetime is an ambition too high. We should explain as MTTF depends on the failure mechanisms, and that failure mechanisms cannot be predicted by any initial measurement. It is the same as asking the expected lifetime of a person based on his/her weigth: there is a statistical relationship in extreme high or low cases, but not in general. My paper, continuing the analogy, proposes a way for distinguishing in that weight the proportion between muscles and fat. This makes a lifetime prevision a little bit easier.

I have distributed along the text, mostly in Introduction and Conclusion, several comments about the range of applicability of the methods.

Round 2

Reviewer 3 Report

The submitted work discusses methods of measurement of optical gain in diode lasers. My opinion concerning it is as follows:

  1. In the beginning of the manuscript it is stated that it is meant for system developers using commercial laser diodes. It is not clear what useful information they may be able to obtain from reading this text, apart from general phrases about the relation between gain and reliability. Specific recommendations should be given to the intended readership of the article as to which of similar laser diodes are more reliable. Advice, recommendations, and/or instructions are necessary that follow from the reported results.
  2. Device reliability is a non-trivial parameter affected by many participating factors (life time, rate of degradation, statistical spread of these parameters across a batch, &c). Vague definition of this parameter compromises the value of the provided analysis and even reduces it to the level of generalities. The Author needs to formulate more specifically his understanding of reliability and list the factually measured parameters behind the experimentally determined reliability factor.
  3. The Introduction informs us that the Author spent several years analysing reliability of laser diodes and is now trying to establish a link between the observed effects (related to reliability) and optical gain/loss. An accentuated conclusion is needed (in the Conclusion and, perhaps, in the Abstract) about the presence of such a link and practical consequences of this conclusion. The Introduction also mentions that this work does not focus on subtle physical effects, thus emphasising the expectation of practical conclusions on the basis of the presented analysis.

If the Author addresses the offered criticism in a further revision of this manuscript, it may be published in Photonics.

Author Response

This is an endless race. I cannot put into a single paper what the scientific community knows since half a century, namely, the role that practical measurements of relevant physical quantities have within a general Reliability framework.

Reliability is made of statistics and of failure physics. The latter addresses the approach, currently known as Physical Reliability Analysis, that started since 1962 with the first international conference on Physics of Failures. Even Wikipedia recalls that "Physics of failure is a technique under the practice of reliability design that leverages the knowledge and understanding of the processes and mechanisms that induce failure to predict reliability and improve product performance". Along that line, conferences as IRPS (https://www.irps.org) in the USA or ESREF (https://esref2021.sciencesconf.org/) in Europe even in their name state that they deal, since decades, with physical reliability.

My paper deals with Physical Reliability, and states this point in its Introduction, and all references are provided for the interested readers.

The reviewer's request, after the 1st review round, of correlating optical gain, measured on a new device  and MTTF  ("One naturally expects to find some dependencies of laser diode life times (how many thousands of hours) upon certain generation parameters of those diodes, but the manuscript gives none") was the negation of half a century of Reliability.

After the 2nd round, my attempts to point out that separating gain from losses IS a new result, and is relevant for Physical Reliability, have been considered "general phrases about the relation between gain and reliability".

I cannot and I want not add further material for discussing the  incompatibility of bathtube and lognormal plots, the false magics of the Weibull
distribution, the problematic extension of the Arrhenius law to non-thermal stresses, the myth of the activation energy, the reason for which the same sets of data lead to MTTF of years or centuries, depending on the non-physical choice of the distribution, and so on. It is known.

Any criticism based on state-of-the-art knowledge will be welcome, and will be considered and meditated in detail. On the contrary, I cannot run this race.